# FMint: Bridging Human Designed and Data Pretrained Models for Differential Equation Foundation Model

## Abstract

The fast simulation of dynamical systems is a key challenge in many scientific and engineering applications, such as weather forecasting, disease control, and drug discovery. With the recent success of deep learning, there is increasing interest in using neural networks to solve differential equations in a data-driven manner. However, existing methods are either limited to specific types of differential equations or require large amounts of data for training. This restricts their practicality in many real-world applications, where data is often scarce or expensive to obtain. To address this, we propose a novel multi-modal foundation model, named **FMint** (**F**oundation **M**odel based on **Init**ialization), to bridge the gap between human-designed and data-driven models for the fast simulation of dynamical systems. Built on a decoder-only transformer architecture with in-context learning, FMint utilizes both numerical and textual data to learn a universal error correction scheme for dynamical systems, using prompted sequences of coarse solutions from traditional solvers. The model is pre-trained on a corpus of 400K ODEs, and we perform extensive experiments on challenging ODEs that exhibit chaotic behavior and of high dimensionality. Our results demonstrate the effectiveness of the proposed model in terms of both accuracy and efficiency compared to classical numerical solvers, highlighting FMint's potential as a general-purpose solver for dynamical systems. Our approach achieves an accuracy improvement of 1 to 2 orders of magnitude over state-of-the-art dynamical system simulators, and delivers a 5X speedup compared to traditional numerical algorithms.

## 1 Introduction

Dynamical systems characterize the evolution of physical states over time. They are fundamental in describing the change of physical states across a wide range of disciplines, including physics (Temam, 2012; Meiss, 2007; Blackmore et al., 2011), chemistry (Tél et al., 2005; Vidal & Pacault, 2012), engineering (Marinca & Herisanu, 2012; Wiggins, 2005; Goebel et al., 2009), and finance (Guegan, 2009; Dong et al., 1996). Typically, these systems are formulated as systems of ordinary differential equations (ODEs):

$$\frac{d\boldsymbol{u}(t)}{dt} = f[\boldsymbol{u}(t)], \quad \boldsymbol{u}(0) = \boldsymbol{c}_0, \tag{1}$$

where $\boldsymbol{c}_0$ denotes the initial condition of the system. To solve these systems numerically, one usually employs a human-designed numerical integration algorithm such as the Euler method or Runge-Kutta methods. These methods can be adapted easily to solve different types of ODEs that share the same format with guaranteed accuracy. The implementation is given as

$$\boldsymbol{u}_{n+1} = \boldsymbol{u}_n + S(f, \boldsymbol{u}_n, \Delta t_n), \quad \boldsymbol{u}_0 = c_0, \quad n = 0, 1, \cdots, \tag{2}$$

where $S$ represents the numerical integration scheme, $\Delta t_n$ is the step size at the $n$-th time step, and $\boldsymbol{u}_n \in \mathbb{R}^n$ is the approximated solution at the cumulative time $\sum_{i=0}^{n} \Delta t_i$.

One obstacle of these human-designed algorithm is the trade-off between accuracy and efficiency. This makes the large-scale simulation using these numerical schemes impossible. In fact, in many

real-world scenarios, high-volume simulation that produces forecasts on a set of initial conditions simultaneously plays a significant role in various applications. For example, simulations of virus propagation during an epidemic given different circumstances are necessary for formulating health regulations (Huang et al., 2023). In these scenarios, it is practical to standardize the time step $\Delta t := \Delta t_1 = \Delta t_2 = \cdots$ across simulations, facilitating batch processing. Yet, this standardization introduces a trade-off between accuracy and efficiency: a larger time step speeds up the simulation at the cost of increased simulation error, while a smaller time step reduces the error but slows down the simulation. Therefore, the long runtime makes these traditional algorithms unsuitable for wide range simulations in many practical situations.

Recently, deep learning methods have demonstrated remarkable success across various scientific domains, including solving partial differential equations (PDEs) (Karniadakis et al., 2021; Wang et al., 2024a), learning operators (Li et al., 2010), and addressing inverse problems (Ongie et al., 2020; Li et al., 2020a; Aggarwal et al., 2018). However, they typically underperform in data-scarce environments and may lack essential domain knowledge. Therefore, we ask an important question:

*Is it possible to combine the best of both worlds, leveraging the efficiency of human-designed algorithms and the accuracy of data-driven methods?*

To address this, we adopt a multi-modal approach to develop a foundation model, **FMint** (**F**oundation **M**odel based on **Init**ialization), a pre-trained foundation model designed to speed up large-scale simulations of dynamical systems with high accuracy via error correction. We integrate human expertise i.e., traditional ODE solvers into modern data-driven methods and adapt the idea of in-context learning to obtain refined solutions based on the initialization of coarse solutions that are computed using human-designed integration method for various differential equations.

In the scientific computing field, most models use solely numerical data as input. However, we ask: Can we integrate other data modalities in scientific computing to enrich the information available? Various characteristic behaviors of dynamical systems can be generalized by textual descriptions, providing more context to the systems. For example, for the Schrödinger equation

$$i\hbar\frac{\partial}{\partial t}\Psi(x,t) = -\frac{\hbar^2}{2m}\nabla^2\Psi(x,t) + V(x)\Psi(x,t), \tag{3}$$

where $\Psi(x,t)$ is the wave function, $\hbar$ is the Planck constant, and $V(x)$ is the potential energy, the possible energy levels of the electron, its spatial distribution, and its interaction with the environment has been analyzed. We hence further incorporate textual modalities in our model that provide guidance to the numerical data.

**Modal 1: Numerical Data** For the numerical modal data, we input the pairs of coarse solutions from traditional numerical solver and the corresponding error correction term to the model. Based on *in-context learning*, given prompted sequences of such pairs, the model is trained to predict the error correction term for the query coarse solution to achieve high accuracy. Such a paradigm allows the model to generalize to various downstream tasks with a short prompt of examples, making it efficient and accurate in a data-scarce environment.

**Modal 2: Textual Data (optional)** FMint can also take textual information as supplemental input to the model. It includes the descriptive information about the equations such as the mathematical expression, or physical meaning of the parameters, etc. We would like to mention here that the textual data is only served as an optional input and is not necessary if not available.

Through extensive experiments involving ODEs with qualitatively different behaviors (periodic, chaotic, etc), we demonstrate the effectiveness of FMint in terms of both accuracy and efficiency over classical numerical solvers and deep learning based methods.

**We summarize our contributions as follows:**

**(1)** To the best of our knowledge, FMint is the first multi-modal foundation model that synthesizes human-designed algorithms and deep learning framework. Back-boned on the decoder-only transformer with in-context learning scheme, FMint achieves competitive accuracy and efficiency in simulating high-dimensional ODEs with qualitatively different behaviors.

**(2)** We obtained 10 to 100 times higher accuracy than state-of-the-art dynamical system simulators, and 5X speedup compared to traditional numerical algorithms with remarkable generalization ability.

## 2 METHODOLOGY

### 2.1 MULTI-MODAL DATA PREPARATION

FMint is a multi-modal foundation modal that bridges human-designed algorithms and data-driven methods. It takes two types of modalities: 1) Numerical Data and, 2) Textual data. The details of the data preparation is summarized below.

**Modality 1: Numerical Data** In solving 1 for large-scale simulations, we consider selecting a numerical integration scheme that utilizes a large time step size. This can be written in stride $k \in \{1, 2, ...\}$ and step size $\Delta t$ that results in desired accuracy, denoted as $k\Delta t$. For illustrative purposes, we consider the Euler method, which yields the following numerical simulation scheme:

$$\hat{\boldsymbol{u}}(t + k\Delta t) = \hat{\boldsymbol{u}}(t) + f[\hat{\boldsymbol{u}}(t)] \cdot k\Delta t. \tag{4}$$

However, solving the dynamical system 1 with numerical scheme 4 and large step size $k\Delta t$ unavoidably causes large simulation errors. From the Taylor expansion

$$\boldsymbol{u}(t + k\Delta t) = \underbrace{\boldsymbol{u}(t) + f[\boldsymbol{u}(t)] \cdot k\Delta t}_{\text{For Euler method}} + \sum_{n=2}^{\infty} \underbrace{\frac{1}{n!} \frac{\mathrm{d}^n}{\mathrm{d}t^n} \boldsymbol{u}(t) \cdot [k\Delta t]^n}_{\text{err}_n(k, \Delta t, \boldsymbol{u}(t))}, \tag{5}$$

we see that the error term $\sum_{n=2}^{\infty} \text{err}_n(k, \Delta t, \boldsymbol{u}(t))$ is non-negligible and this limits the fast simulation of real-world dynamical systems. We therefore consider building a corrector foundation model that approximates $\sum_{n=2}^{\infty} \text{err}_n$ for various dynamical systems. We call solutions obtained by vanilla numerical integration schemes 4 with time step $k\Delta t$ as "coarse solutions". With coarse solutions as an initialization, our goal is to produce highly accurate solution with fast inference time on a diverse set of dynamical systems, i.e.,

$$\hat{\boldsymbol{u}}_{k(n+1)} = \hat{\boldsymbol{u}}_{kn} + S\left(f, \hat{\boldsymbol{u}}_{kn}, k\Delta t\right) + \text{FMint}\left(\hat{\boldsymbol{u}}_{kn}; \Theta\right), \quad \hat{\boldsymbol{u}}_0 = \boldsymbol{c}_0, \quad n = 0, 1, \cdots, \tag{6}$$

where $\Theta$ represents all the model parameters. We designed our model using a decoder-only transformer backbone (Vaswani et al., 2017). The model is trained to perform in-context learning such that it predicts the error correction term in examples based on previous demonstrations. The training is done in a similar manner to the next-token-prediction scheme.

**Training Data Preparation.** We construct FMint to learn the corrector from multiple demos from the same ODE system, each consists of coarse solutions and their corresponding correction term. To prepare for the training data, for $i$-th ODE equation, we first simulate using fine step size $\Delta t$ and obtain ODE $\{\boldsymbol{u}_j^i\}_{j=1}^{kn}$ where $\boldsymbol{u}_j^i$ represents the fine-grained solution for $i$-th ODE system at time step $j\Delta t$. Then using coarse step size $k\Delta t$, we generate ODE results $\{\hat{\boldsymbol{u}}_{kj}^i\}_{j=1}^{n}$ where we denote $\hat{\boldsymbol{u}}_{kj}^i$ the coarse solution for $i$-th ODE equation at time step $kj\Delta t$ with predefined stride $k$. The corresponding error correction term for each coarse solutions are computed from the difference

$$\text{err}_{\hat{\boldsymbol{u}}_{kj}} = \boldsymbol{u}_{k(j+1)} - \hat{\boldsymbol{u}}_{kj} - S\left(f, \hat{\boldsymbol{u}}_{kj}, k\Delta t\right). \tag{7}$$

One pair of coarse solutions $\hat{\boldsymbol{u}}^i = \{\hat{\boldsymbol{u}}_{kj}^i\}_{j=1}^{n}$ and error term $\text{err}^i = \{\text{err}_{\hat{\boldsymbol{u}}_{kj}^i}\}_{j=1}^{n}$ composes one *demo*. The model takes a collection of demos of size $d$, a query data sequence $\hat{\boldsymbol{u}}^t$ and outputs an error correction term $\text{err}^t$ for the query data

$$\left\{\{\hat{\boldsymbol{u}}^1, \text{err}^1\}, \{\hat{\boldsymbol{u}}^2, \text{err}^2\}, \ldots, \{\hat{\boldsymbol{u}}^d, \text{err}^d\}, \hat{\boldsymbol{u}}^t\right\} \rightarrow \text{err}^t. \tag{8}$$

All the demo information will be tokenized before passed into the FMint model. We describe the tokenization in detail in Subsection 2.2.

**Pretraining Data.** We pretrain the FMint model with four types of ODEs that exhibit qualitatively different characteristics to cover a wide range of diversity in data:

*(1) Lorenz model.* The Lorenz system is a 3D system, well-known for its chaotic behavior, originally developed to model atmospheric convection. It is governed by three coupled, nonlinear differential equations.

*(2) Damped oscillator.* A damped oscillator is a 2nd order system that describes a system where the motion of the oscillator is subject to a force that reduces its amplitude over time. This leads to a

decay in oscillation.

*(3) Van der Pol oscillator.* The Van der Pol oscillator is a 2nd-order nonlinear oscillator. It can sustain oscillations indefinitely, known as limit cycles.

*(4) Lotka-Volterra.* The Lotka-Volterra system is a 2D system that describes the nonlinear interaction between two species: the prey and the predator.

**Modality 2: Textual Data.** For the optional multi-modal training, we produced 30 descriptions per ODE as supplemental textual data for training and testing. These data contains the mathematical expression, exact parameters used for each ODEs, or behaviors under different parameter ranges. These data are generated with the help of GPT-4. We provide the details of the textual data generation in Appendix A.5 and provide some examples listed in Appendix A.6.

## 2.2 Model Design and Modality Fusion

In this subsection, we describe the model architecture and the fusion of modalities.

Table 1: Input tokens for a 2D ODE demo.

| key | Coarse solution | | | Error term | | | Query | | |
|---|---|---|---|---|---|---|---|---|---|
| | 0 | ... | $t_n$ | 0 | ... | $t_n$ | 0 | ... | $t_q$ |
| value | $\hat{u}(0)$ | ... | $\hat{u}(t_n)$ | $\mathrm{err}_{\hat{u}}(0)$ | ... | $\mathrm{err}_{\hat{u}}(t_n)$ | $\hat{u}^q(0)$ | ... | $\hat{u}^q(t_q)$ |
| | $\hat{v}(0)$ | ... | $\hat{v}(t_n)$ | $\mathrm{err}_{\hat{v}}(0)$ | ... | $\mathrm{err}_{\hat{v}}(t_n)$ | $\hat{v}^q(0)$ | ... | $\hat{v}^q(t_q)$ |

**Tokenization** FMint processes two modalities of data: numerical and textual. Like other language models, FMint requires tokenization of the input data before passing them into the transformer. The numerical data can be classified into three categories: coarse solutions, error corrections, and query solutions. Query solutions consists of the coarse solution of the query trajectory. As previously discussed, FMint employs in-context learning using pairs of coarse solutions and their associated error corrections to predict the error corrections for the query locations. As shown in Table 1, each column represents a token, with each token belonging to one of the three categories. To handle these three categories, we utilize a shared embedding layer that maps them into embedding vectors. Additionally, a learnable positional embedding is appended to the embedding vectors of each category. Notably, the positional embeddings are consistent within each data category but differ across the categories. For tokenizing textual data, we employ a separate embedding layer, and the resulting embedding vectors are appended with positional embeddings.

**Model architecture.** Similar to language models, FMint is a decoder-only transformer model, where the key design aspect lies in the appropriate masking of the attention mechanism. The mask must satisfy the following requirements: (1) When predicting query locations, the model should be invariant to the order of queries, allowing predictions to be made independently and in parallel. (2) When predicting queries in the current example, the query tokens must have access to the tokens of coarse solutions and error corrections from both previous and the current examples. To effectively manage these constraints, we use a specialized masking technique introduced by Yang et al. (2023b). In particular, the mask is designed such that when predicting the current queried error correction, the model "sees" the prompt, all previous coarse solutions with error terms and queries with the queried error terms, but not the current queried error term. After passing through multiple attention blocks, FMint is connected to a head layer (multi-layer perceptron), which outputs the error correction predictions for the query locations of the system. The architecture of FMint is shown in Figure 1.

**Training and inference.** We pretrain FMint on 400K ODE simulation data using an in-context learning scheme. During training, ODE trajectories with the same parameters but different initial conditions are passed to the model as demo pairs. The query tokens share the same time variables as the error correction tokens, but their values are set to 0. The training loss function is the mean squared error (MSE) 9, which minimizes the difference between the predicted error corrections for the query data and the ground truth error. Thanks to the mask design, the model outputs predictions in a language generation-like manner, using information from both the current and previous examples.

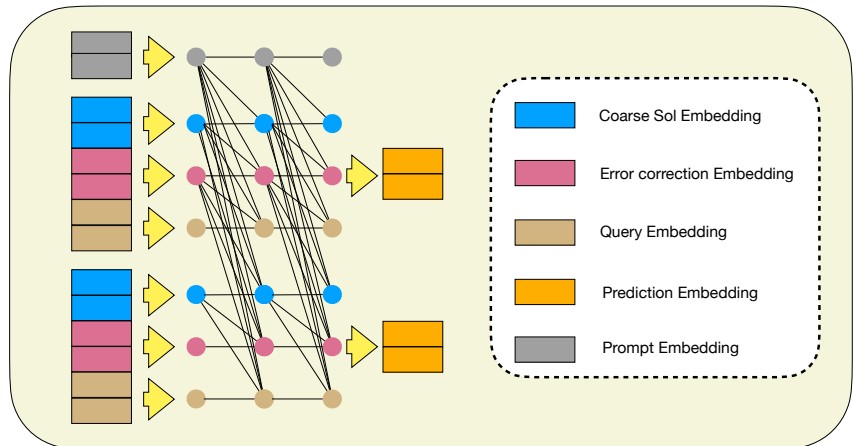

Figure 1: Work flow of FMint model.

$$\text{MSE} = \frac{1}{N} \sum_{i=1}^{N} \|\tilde{\text{err}}^i - \text{err}^i\|_2 \tag{9}$$

During inference, we provide FMint with a few demo pairs of coarse solutions and corresponding error corrections. Once given the query locations, FMint accurately predicts the error corrections, achieving high accuracy even from coarse, low-accuracy simulation data. The generation of query locations is highly flexible and order-invariant, allowing users to perform data interpolation. Notably, for relatively simple dynamical systems, FMint can perform zero-shot or few-shot learning. For more complex systems, FMint still achieves high accuracy with minimal fine-tuning, as shown in the numerical results section 3.

## 3  EXPERIMENTAL RESULTS

In this section, we conduct extensive experiments to evaluate the effectiveness of FMint and compare its results with SOTA baslines (Chen et al., 2018; Huang et al., 2023; Yang et al., 2023a) on a wide range of ODEs from 1D to 3D. We aim to answer the following research questions:

**RQ1.** How does FMint perform compared to baseline models in terms of accuracy and efficiency.

**RQ2.** How does FMint adapt to ODEs' different behavior under various data generation coefficients.

**RQ3.** Whether the optional textual data improves the performance of FMint.

### 3.1  BASIC SET-UP

**Pretraining data preparation.** The pretraining data consists of 400K ODEs that are commonly observed in important applications in engineering and science: (1) Lorenz model, (2) Damped oscillator, (3) Van der Pol oscillator, and (4) Lotka-Volterra (LV) dynamics. For each ODE system, we created 1000 variations with different parameters and for each variation, we produce 100 trajectories with different initial conditions. Consequently, our dataset comprises trajectories of 100,000 ODEs for each dynamical system, differentiated by varying coefficients and initial conditions.

We have included more details on these ODE systems in Appendix A.1. For all four ODE systems, the coefficients' range are provided in Table 8; the time step size $\Delta t$, the value of strides $k$, the range of initial conditions (IC), and the numerical integration scheme used for data generation are summarized in Table 7.

**Implementation details.** As a decoder-only transformer model, FMint is configured with approximately 15.8M parameters. The model features six heads for multi-head attention, with an input/output dimension of 256 for each layer. The embedding vector dimensions for the coarse solution, error correction, and query are set to 256, while the hidden dimension of the feed-forward networks is set to 1024. Pretraining is conducted on a NVIDIA A100 GPU with 80 GB of memory

and finetuning is conducted on a NVIDIA A6000 GPU with 48 GB of memory. We use AdamW optimizer with a warmup-cosine-decay schedule, with peak learning rate 1e-4 and 60 training epochs. The Adam $\beta_1$ and Adam $\beta_2$ are 0.9 and 0.999, respectively, and the weight decay is set to 1e-4. Demo number for training and testing is five.

**Baselines and tasks.** We compare FMint with three baseline models: Neural ODE (Chen et al., 2018), NeurVec (Huang et al., 2023), and In-Context Operator Networks (ICON-LM) (Yang et al., 2023b). Neural ODEs model continuous-time dynamics by parameterizing the derivative of the hidden state with a neural network. This approach turns the forward pass into solving an initial value problem, offering a memory-efficient way to capture temporal patterns in data. NeurVec is a deep learning-based corrector aimed to compensate for integration errors and enable larger time step sizes in simulations. ICON-LM is a foundation model for operator learning, achieving better accuracy and data efficiency than Fourier Neural Operator (Li et al., 2020b) and DeepONet (Lu et al., 2019).

Among them, ICON-LM is a multi-task model trained on a large collection of examples while both Neural ODE and NeurVec fit one neural network per example. The configuration and training details of Neural ODE and NeurVec are provided in Appendix A.3 and A.4, while the settings for ICON-LM follow those outlined in the original work (Yang et al., 2023b).

**Evaluation metrics.** We use the mean absolute errors (MAE) and root mean square errors (RMSE) compared to fine-grained ODE solutions as the evaluation metric:

$$\text{MAE} = \frac{1}{N} \sum_{i=1}^{N} \|\tilde{\boldsymbol{u}}^i - \boldsymbol{u}^i\|_2, \quad \text{RMSE} = \sqrt{\frac{1}{N} \sum_{i=1}^{N} \|\tilde{\boldsymbol{u}}^i - \boldsymbol{u}^i\|_2}, \tag{10}$$

where $\tilde{\boldsymbol{u}}^i$ is the predicted ODE solution for the $i$-th equation. For FMint, it can be computed via error correction $\tilde{\boldsymbol{u}}_k^i = \hat{\boldsymbol{u}}_k^i + \hat{\text{err}}^i$ such that $\hat{\boldsymbol{u}}_k^i$ is the coarse solution of the $i$-th equation, and $\hat{\text{err}}^i$ is the model output by FMint.

### 3.2 RQ1. FMINT V.S. BASELINS MODELS

Here we show that FMint outperforms baseline methods on both in-distribution and out-of-distribution data in terms of accuracy and efficiency.

**In-distribution ODEs**

*Accuracy.* We evaluate FMint on the test split of the pretraining dataset. This contains ODEs from the same ODE families with the same parameter range, but with different random parameters within the range and different initial conditions. For each ODE system, we finetuned our model for 1000 or 2000 epochs with trajectories of size 100. Details on the testing coefficient range and finetuning epochs are included in Table 8.

Table 2: Performance for in-distribution ODEs via MAE and RMSE (lower is better)

| Methods | MAE | | | | RMSE | | | |
|---|---|---|---|---|---|---|---|---|
| | Lorenz | Damped Osci | Van der Pol | LV | Lorenz | Damped Osci | Van der Pol | LV |
| ICON-LM | 2.87 | 0.095 | 1.73 | 5.74 | 4.98 | 0.14 | 2.03 | 9.07 |
| Neural ODE | 17.9017 | 0.0865 | 1.8476 | 6.3497 | 22.6992 | 0.3907 | 2.2656 | 9.2982 |
| NeurVec | 6.6465 | 0.0919 | 1.7583 | 4.9032 | 10.5565 | 0.4499 | 2.0390 | 8.5570 |
| **FMint** | **0.159** | **0.0044** | **0.00878** | **0.0304** | **0.33** | **0.00695** | **0.0119** | **0.0433** |

MAE and RMSE results are shown in Table 2 for all in-distribution ODE families with the best result in bold and the second best with underline. Both metrics are averaged over 25 ODEs with initial conditions from the same family and the number of demos is five during the inference stage. Example visualization of FMint's performance on Lorenz and Van der Pol is shown in Figure (2a) (2b). The fine-grained solution $u_j$ is labeled as *Fine ode*, the coarse solution $\hat{u}_{kj}$ is labeled as *coarse ode* and FMint result is labeled as *FMint ode*. For visualization of FMint on all tested ODEs, see Appendix A.7.

We observe that FMint achieves an accuracy that is at least an order of magnitude higher than all baseline models. For the relatively simple model of damped oscillators, we observe that all models exhibit relatively small errors, with FMint achieving the highest accuracy. However, for the Lorenz

model, where the dynamics are highly chaotic, Neural ODE completely fails to capture the dynamics at large time steps. This underscores the importance of human-designed algorithms in stabilizing the system, especially when the initial conditions vary. NeurVec performs better but remains unsatisfactory, while ICON-LM achieves the best results among the baselines. These observations suggest that pretraining on a diverse dataset is crucial; otherwise, variations in the initial conditions will cause the model to fail, which explains NeurVec's weaker performance in this case. FMint achieves the best performance due to both its pretraining and the combination of data-driven and human-designed algorithms.

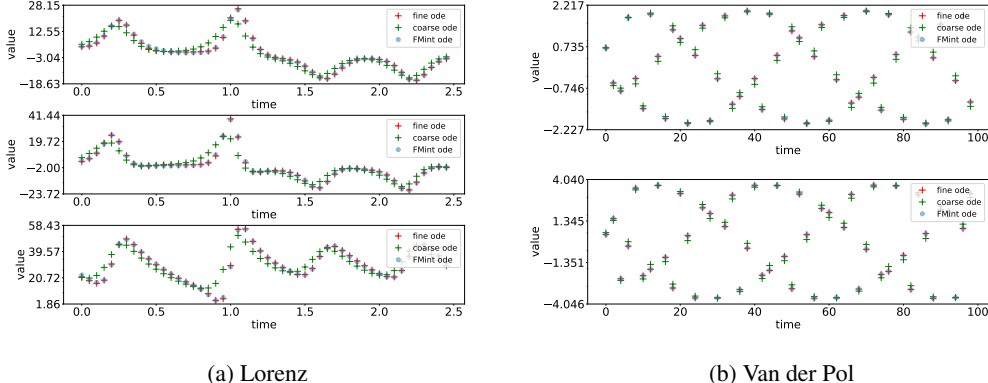

(a) Lorenz  (b) Van der Pol

*Efficiency.* We further examined the runtime of FMint in comparison with fine solution generation using RK4. The test is conducted on Lotka-Volterra system with 500 equations. To display the runtime better, we use the runtime for obtaining coarse solutions using RK4 as one unit, and we report the result in Figure 3. FMint is able to attain results with comparable accuracy to the fine solutions (RK-FINE) using less than 20% of its time.

**Out-of-distribution (OOD) ODEs**

To further demonstrate the performance of FMint on unseen ODEs from different ODE families, we use data simulated from the following dynamical systems: driven-damped pendulum (2D), falling object with air resistence (2D), FitzHugh-Nagumo systems (2D), Pendulum under gravity (2D), and the Rössler dynamics (3D). These test ODEs are qualitatively different from the training data, thus convincing to validate FMint's capability as a foundation model for dynamical systems. For more details about the OOD ODEs, see Appendix A.2. We evaluate the transfer performance of FMint using trajectories of size $N \in \{25, 50, 200, 1000\}$. We finetune the model for 1000 to 2000 iterations on the new data of size $N$ and report the RMSE and MAE on the test data. Details on the coefficient range and finetuning epochs are included in Table 8. As a comparison, RMSE and MAE are computed for NeurVec and Neural ODE using training set of size 50K. The results are shown in Table 3

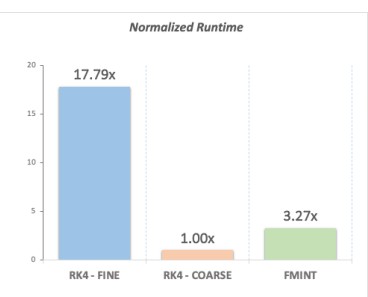

Figure 3: Normalized runtime for Lotka-Volterra of reference solution (RK4-FINE) coarse solution (RK4-COARSE) and FMint.

We observe that with only 25 training trajectories, FMint is able to outperform NeurVec and Neural ODE trained on data with sample size 50000. Although increasing the training sample size may improve the accuracy of FMint, finetuning with small sample size still gives us robust results. This shows that FMint has great potential for large-scale simulations in many real-world scenarios where data collection is expensive.

**MAE v.s. training epochs.** To better examine what factors may affect FMint's performance for unseen ODEs other than training sample size, we fine tuned our model on Lorenz equation with unseen coefficient range $\rho \in (100, 150)$. The system exhibits more complex chaotic dynamics with larger attractor regions and more intricate patterns under this range. We plotted the MAE on the testing split of trajectories of size 100 with respect to finetuning epochs in Figure 4 with one standard deviation. As expected, we see that MAE decreases as training epochs increases but starts to saturate after a certain point.

Table 3: Performance of FMint on unseen ODEs in MAE.

| Method | #Samples | Unseen ODE | | | | |
|---|---|---|---|---|---|---|
| | | Driven damped | Falling | Fitzhugh Nagumo | Pendulum | Rössler |
| FMint | 25 | 1.06e-3 | 2.71e-3 | 9.53e-3 | 1.10e-3 | 1.61e-3 |
| | 50 | 6.77e-4 | 2.99e-3 | 9.79e-3 | 1.26e-3 | 1.61e-3 |
| | 200 | 9.32e-4 | 2.88e-3 | 7.67e-3 | 1.15e-3 | 1.53e-3 |
| | 1000 | 9.02e-4 | 2.51e-3 | 6.01e-3 | 1.25e-3 | 1.43e-3 |
| NeurVec | 50000 | 0.0538 | 0.7068 | 0.2782 | 0.0666 | 0.0084 |
| Neural ODE | 50000 | 0.5592 | 0.9743 | 0.9314 | 0.8519 | 0.0795 |

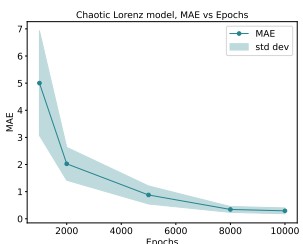

Figure 4: MAE v.s. fine-tuning epochs for Lorenz equation.

## 3.3 RQ2. UNIFORMITY OF FMINT'S PERFORMANCE

**On dynamical systems with qualitatively different behaviors.**

For the same dynamical systems, different parameter ranges can lead to vastly different behaviors. For example, the Lorenz system is highly sensitive to initial conditions and parameter values, leading to chaotic behavior under certain conditions. This is challenging for traditional numerical solvers to simulate with large time steps, since even small errors introduced by large time steps can result in significant deviations over time due to the sensitivity to initial conditions. In addition, chaotic systems exhibit fine-scale behaviors, such as oscillations, sharp changes, or bifurcations. Large time steps may skip over these critical behaviors, missing important features like turning points, sharp gradients, or periodicity.

For example, in Lorenz system, when $\rho \in [13, 24.74)$, the system exhibits converging oscillatory behavior toward either of the two fixed points. But when $\rho \in [24.74, 100)$, the system exhibits chaotic behavior and sensitive to initial conditions. As $\rho$ increases further e.g., $\rho > 100$, the chaotic behavior becomes more complex. Figure 5 shows trajectories generated with $\sigma = 12.69, \beta = 2.59, \rho = 24.33$ under two initial conditions $[1, 1, 1]$ and $[20, 20, 20]$ (top) and trajectories generated with $\sigma = 12.69, \beta = 2.59, \rho = 78.06$ under two initial conditions $[1, 1, 1]$ and $[20, 20, 20]$ (bottom). Both trajectories are simulated for 10000 steps with $\Delta t = 0.004$.

Table 4: Performance of FMint on systems exhibiting qualitatively different behaviors, measured in MAE and RMSE. Driven damped pendulum and FitzHugh-Nagumo are shorten to DDP and FHN.

| Name | MAE | RMSE |
|---|---|---|
| Lorenz oscillatory | 0.067 | 0.114 |
| Lorenz chaotic | 0.129 | 0.209 |
| Lorenz chaotic (complex) | 0.29 | 0.614 |
| DDP underdamped | 1.35e-3 | 1.73e-3 |
| DDP overdamped | 5.66e-4 | 8.47e-4 |
| DDP chaotic | 1.39e-3 | 1.81e-3 |
| FHN resting | 1.63e-3 | 2.71e-3 |
| FHN spikes | 1.9e-3 | 2.97e-3 |
| FHN bursting | 6.95e-4 | 1.45e-3 |
| FHN oscillatory | 8.18e-4 | 1.60e-3 |
| Rössler periodic | 1.43e-3 | 2.36e-3 |
| Rössler chaotic | 1.44e-3 | 2.34e-3 |
| Rössler hyperchaos | 1.60e-3 | 2.70e-3 |

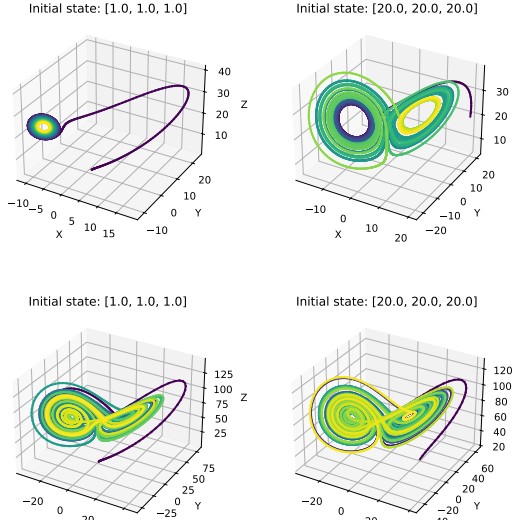

Figure 5: Figures showing Lorenz attractors' behavior under two different sets of parameters.

Here we conduct extensive experiments of FMint in such cases to test its generalization ability. Conretely, we test on three of Lorenz, three of Driven damped pendulum (DDP), four of FitzHugh-Nagumo (FHN) and three of Rössler that exhibits various behaviors under different range of coeffi-

cients. Results are reported in Table 4. For more details on the coefficient range for each behavior category, see Appendix A.8.

**On pretraining ODEs with different strides.**

In addition, we test our pretrained model on test data simulated using different strides $k$ than the pretraining data. This examines how adaptable FMint is to handle realistic circumstances in which the coarse solution simulation varies during the inference stage. For consistency over various families, we generate test examples with new stride values proportional to the training strides: $\alpha k, \alpha = \{0.5, 2.0\}$.

For each ODE system, we finetuned our model for 1000 or 2000 epochs with trajectories of size 100. Table 5 reports MAE and RMSE of FMint for smaller or larger strides $k$. We observe that the accuracy of all four ODE families is consistent with the accuracy of the training strides.

Table 5: MAE and RMSE of FMint under strides differs from pretraining data.

| Name | $k$ | MAE | RMSE | $k$ | MAE | RMSE |
|---|---|---|---|---|---|---|
| Lorenz model | 50 | 4.5e-2 | 9.4e-2 | 200 | 3.15e-2 | 4.65e-2 |
| Damped oscillator | 50 | 3.58e-3 | 5.51e-3 | 200 | 4.03e-3 | 6.54e-3 |
| Van der Pol | 5 | 1.6e-3 | 2.11e-3 | 20 | 2.76e-3 | 3.60e-3 |
| Lotka-Volterra | 50 | 6.51e-3 | 1.01e-2 | 200 | 1.86e-2 | 2.64e-2 |

## 3.4 RQ3. MULTI-MODAL FMINT LEARNING

**Performance of FMint with the supplemental textual data.**

We further investigated the performance of FMint with the supplemental textual data for both pretraining and out-of-distribution ODEs. We first pretrained the model on the pretraining data but with prepared textual data for each ODEs. Then, the model is further finetuned on each ODEs for 1000 or 2000 epochs on trajectories of size 100. For consistency, we used the exact same numerical dataset for both pretraining FMint without prompts and finetuning as in Section 3.2. For details on the generation of the textual data, see Appendix A.5.

Table 6 shows the MAE and RMSE of FMint with and without the prompts. We observe that FMint, combined with prompts, enhances performance in challenging dynamical systems such as the Lorenz, Van der Pol, Lotka-Volterra, and Rössler systems. However, in relatively simpler cases like the Damped Oscillator and Pendulum under gravity, the inclusion of prompts does not improve performance, remaining comparable to baseline FMint results. This suggests that the multi-modal foundation model is particularly advantageous when simulating complex and challenging systems.

Table 6: Compairson of FMint and FMint with textual data in MAE and RMSE

| | MAE | | | | | | | | |
|---|---|---|---|---|---|---|---|---|---|
| Methods | Lorenz | Damped Osci | Van der Pol | LV | Driven damped | Falling | Fitzhugh Nagumo | Pendulum | Rössler |
| FMint | 0.159 | 4.4e-3 | 8.78e-3 | 3.04e-2 | 6.77e-4 | 2.99e-3 | 9.79e-3 | 1.26e-3 | 1.61e-3 |
| FMint + prompt | 0.109 | 4.52e-3 | 4.29e-3 | 1.76e-2 | 8.12e-4 | 2.66e-3 | 9.90e-3 | 2.84e-3 | 1.40e-3 |
| | RMSE | | | | | | | | |
| Methods | Lorenz | Damped Osci | Van der Pol | LV | Driven damped | Falling | Fitzhugh Nagumo | Pendulum | Rössler |
| FMint | 0.33 | 6.95e-3 | 1.19e-2 | 4.33e-2 | 9.68e-4 | 5.26e-3 | 5.8e-2 | 1.7e-3 | 2.56e-3 |
| FMint + prompt | 0.224 | 6.97e-3 | 5.87e-3 | 2.54e-2 | 1.17e-3 | 4.35e-3 | 9.90e-3 | 2.78e-3 | 2.19e-3 |

## 4 RELATED WORK

**Neural network for dynamical systems.** In recent years, neural network-based solvers have been widely applied to scientific problems such as solving ODEs, PDEs, operator learning, and inverse problems. One common approach parameterizes PDE solutions using feed-forward neural networks (Han et al., 2018; 2017; Karniadakis et al., 2021; Cui et al., 2024; De Florio et al., 2023; Sirignano & Spiliopoulos, 2018; Yu et al., 2018; Yuan et al., 2024; Wang et al., 2024b), where physical laws are enforced via hard or soft constraints in the loss function. The Finite Expression Method (FEX) (Liang & Yang, 2022; Song et al., 2023; 2024) offers an interpretable alternative by representing PDE solutions in computer algebra, capturing solution structure with high accuracy. Neural operator

learning (Li et al., 2010; Ong et al., 2022; Cao, 2021; Li et al., 2022; Zhang et al., 2021; Lu et al., 2021) maps varying parameters or initial conditions to solutions, achieving discretization invariance but requiring large amounts of high-quality data and lacking generalization to unseen distributions.

Recent research has explored integrating traditional numerical algorithms with deep learning to improve the accuracy of dynamical system simulations (Guo et al., 2022; Huang et al., 2023). For instance, NeurVec (Huang et al., 2023) enables rapid ODE simulations with large time steps, achieving decent accuracy on several classic systems. However, its limited generalization to out-of-distribution systems restricts its practicality for large-scale real-world simulations.

**Foundation model in scientific machine learning.**

Large language models like GPT-4 (Achiam et al., 2023), DALL-E (Ramesh et al., 2021), and Llama (Touvron et al., 2023) have achieved remarkable success across various domains (Thirunavukarasu et al., 2023; Devlin et al., 2018; Sun et al., 2024; Qin et al., 2023; Hu et al., 2024; Li et al., 2023; Bi et al., 2024; Jiang et al., 2024), including text-to-visual generation (Ji et al., 2024; Ji & Liu, 2024) and information retrieval (Kang et al., 2024). These models leverage extensive pre-training and adapt to downstream tasks via zero-shot or few-shot learning (Wang et al., 2023; Yu et al., 2023), demonstrating impressive transfer learning capabilities. Inspired by these advances, foundation models have gained traction in scientific machine learning. For example, Subramanian et al. (2024) explored the Fourier Neural Operator (FNO) (Li et al., 2010) for solving classical PDEs, while the Unified PDE Solver (UPS) (Shen et al., 2024) extended this approach to 1D and 2D PDEs using pre-trained models. Additionally, McCabe et al. (2023) embedded PDEs with varying properties into a shared space, and Rahman et al. (2024) enhanced attention mechanisms for handling PDEs with diverse dimensions.

A growing area in scientific machine learning is in-context learning (Dong et al., 2022; Xie et al., 2021; Olsson et al., 2022; Wei et al., 2022; Xu et al., 2024), where models are prompted with example pairs and trained to predict new queries based on observed patterns. The In-context Operator Network (ICON) (Yang et al., 2023a; Yang & Osher, 2024; Yang et al., 2023b) applies this approach to operator learning by leveraging example pairs with varying PDE parameters and solutions to predict solutions for new query data.

## 5   CONCLUSION

In this paper, we presented FMint, a novel multi-modal foundation model that speeds up large-scale simulations of dynamical systems. Based on the architecture of decoder-only transformer, FMint takes textual and numerical data to deliver high-accuracy simulation based on initial solution from traditional numerical solvers. FMint incorporates the in-context learning for a universal error corrector for ODEs from given prompted sequences of coarse initialized solutions. It is pre-trained using a diverse set of ODE families with qualitatively different behaviors.

We show that FMint achieves a significant improvement in accuracy over state-of-the-art dynamical system simulators and accelerates traditional integration schemes. In comparison to direct ODE solvers, we recognize the importance of integrating the strengths of human-designed algorithms and data-driven methods for the simulation of dynamical systems. The in-context learning scheme enables it to effectively interpolate to arbitrary time points, enhancing its versatility in handling temporal dynamics. Furthermore, we propose a multi-modal foundation model perspective to address scientific computing problems, challenging the mainstream numerical-data-only approach in the field. Given FMint's performance, it shows promise for scaling up to simulate even more complex dynamics in real-world applications.

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

# A APPENDIX

## A.1 PRETRAINING ODE DETAILS

- The Lorenz Model is described by

$$\begin{cases} \dfrac{dx}{dt} = \sigma(y - x), \\[2mm] \dfrac{dy}{dt} = x(\rho - z) - y, \\[2mm] \dfrac{dz}{dt} = xy - \beta z, \end{cases}$$

where $x$, $y$, and $z$ represent the convection rate, horizontal temperature variation, and vertical temperature variation, respectively. The Prandtl number, $\sigma$, is a dimensionless quantity indicating the ratio of momentum diffusivity to thermal diffusivity, while the Rayleigh number, $\rho$, quantifies the temperature gradient driving convection. The parameter $\beta$ represents the geometric aspect ratio of the convective cells in the model.

- Damped oscillator equation is given by:

$$\frac{d^2x}{dt^2} + 2\zeta\omega\frac{dx}{dt} + \omega^2 x = 0,$$

where $\zeta$ is the damping ratio, and $\omega$ is the natural frequency.

- The Van der Pol Oscillator is given by the following equation:

$$\frac{d^2x}{dt^2} - \mu(1 - x^2)\frac{dx}{dt} + x = 0,$$

where $\mu$ controls the degree of nonlinearity and damping.

- The Lotka-Volterra system is given by the following equations:

$$\begin{cases} \dfrac{dx}{dt} = \alpha x - \beta xy, \\[2mm] \dfrac{dy}{dt} = \delta xy - \gamma y, \end{cases}$$

where $x$ is the number of prey, $y$ is the number of predator, $\alpha$ is the natural growing rate of prey in the absense of predators, $\beta$ is the natural dying rate of prey due to predation, $\gamma$ is the natural dying rate of predators in the absence of prey, and $\delta$ is the rate at which predators increase by consuming prey.

## A.2 TESTING ODE DETAILS

- Rössler attractor is a system of three nonlinear ODEs, which exhibits chaotic dynamics. The equation is given by

$$\begin{cases} \dfrac{dx}{dt} = -y - z, \\[2mm] \dfrac{dy}{dt} = x + ay, \\[2mm] \dfrac{dz}{dt} = b + z(x - c), \end{cases}$$

where $x, y, z$ are state variables, $a, b, c$ are parameters that control the behavior of the system.

- FitzHugh-Nagumo model is used for neuron activity dynamics. It captures the essential features of neuronal excitability, including the generation and propagation of action potentials. The equation is given by

$$\begin{cases} \dfrac{dv}{dt} = v - \dfrac{v^3}{3} - w + I, \\[2mm] \dfrac{dw}{dt} = \epsilon(v + a - bw), \end{cases}$$

where $v$ is membrane potential, $w$ is a recovery variable, $I$ is an external stimulus current. Parameter ranges are listed as follows: $\epsilon$, $a$, $b$ and $I$.

- This equation describes the dynamics of the falling object with air resistance:

$$\frac{d^2x}{dt^2} = g - c\frac{dx}{dt},$$

where $g$ represents the gravitational constant, and $c$ is the coefficient of drag.

- This equation describes how the pendulum swings under the influence of gravity and damping:

$$\frac{d^2x}{dt^2} = -\frac{g}{l}\sin(x) - b\frac{dx}{dt},$$

where $g$ represents the gravitational constant, and $l$ is the length of pendulum, and $b$ is the damping coefficient.

- This is a classic problem in the study of dynamical systems and chaotic behavior, particularly under the influence of non-linear restoring forces and external driving forces. It is written as:

$$\frac{d^2\theta}{dt^2} + b\frac{d\theta}{dt} + c\sin(\theta) = A\cos(\omega t),$$

where $\theta$ represents the angular displacement from the vertical, $b$ is the damping coefficient, $c$ is the gravitational constant times the length of the pendulum, $A$ is the amplitude, and $\omega$ is the angular frequency of the drive.

Table 7: Data generation setup

| Name | $k$ | $\Delta t$ | IC (1st dim) | IC (2nd dim) | IC (3rd dim) | Integration |
|---|---|---|---|---|---|---|
| Lotka-Volterra | 100 | 0.001 | $(10, 100)$ | $(5, 10)$ | NA | RK4 |
| Van der Pol | 10 | 0.01 | $(-1.0, 1.0)$ | $(-0.5, 0.5)$ | NA | RK4 |
| Damped Osci. | 100 | 0.0001 | $(-2.0, 2.0)$ | $(-0.1, 0.1)$ | NA | RK4 |
| Lorenz | 100 | 0.0001 | $(-5, 5)$ | $(-5, 5)$ | $(0, 25)$ | RK4 |
| Fitzhugh Nagumo | 100 | 0.005 | $(-1.0, 1.0)$ | $(-0.5, 0.5)$ | NA | RK4 |
| Falling object | 20 | 0.01 | $(0, 100)$ | $(0, 2)$ | NA | RK4 |
| Pendulum gravity | 20 | 0.01 | $(0, \frac{\pi}{4})$ | $(-\frac{\pi}{4}, \frac{\pi}{4})$ | NA | RK4 |
| Driven damped pendulum | 20 | 0.01 | $(-\frac{\pi}{4}, \frac{\pi}{4})$ | $(-0.5, 0.5)$ | NA | RK4 |
| Rössler | 100 | 0.001 | $(-1, 1)$ | $(-1, 1)$ | $(-1, 1)$ | RK4 |

## A.3 NEURAL ODE IMPLEMENTATION DETAILS

**Neural ODE Architecture.** The neural ODE model employed in our experiments consists of a fully connected feedforward neural network with the following architecture:

- **Input layer:** The state dimension of the system (varies per system).
- **Hidden layers:** Two hidden layers, each with 1024 neurons and Tanh activation functions.
- **Output layer:** The derivative of the state with respect to time (same dimension as the input state).

The model is formally described as:

$$f_\theta(t, \boldsymbol{x}) = \text{Linear}(\text{Tanh}(\text{Linear}(\boldsymbol{x}))).$$

**Neural ODE Training Details.** We trained the neural ODE models on various dynamical systems using fine time step data and evaluated them on coarse time step data. The specific training parameters were as follows:

- **Learning Rate:** 0.001
- **Optimizer:** Adam
- **Learning Rate Decay:** StepLR with a decay factor of 0.5 every 20 epochs
- **Batch Size:** 500
- **Number of Epochs:** 100
- **Loss Function:** Mean Squared Error (MSE)

Table 8: Coefficient range for pre-training and testing and epochs for fine-tuning

| Name | pretraining | testing | finetuning epochs |
|---|---|---|---|
| Lotka-Volterra | $\alpha \in [0.1, 1.0]$ $\beta \in [0.01, 0.1]$ $\gamma \in [0.1, 1.0]$ $\delta \in [0.01, 0.1]$ | $\alpha \in [0.05, 1.2]$ $\beta \in [0.005, 0.15]$ $\gamma \in [0.05, 1.2]$ $\delta \in [0.005, 0.15]$ | 2000 |
| Van der Pol | $\mu \in [0.1, 5]$ | $\mu \in [0.05, 10]$ | 1000 |
| Damped Osci. | $\zeta \in [0.1, 2.0]$ $\omega \in [0.5, 5.0]$ | $\zeta \in [0.1, 7.0]$ $\omega \in [0.5, 8.0]$ | 1000 |
| Lorenz | $\sigma \in [8, 12]$ $\rho \in [20, 30]$ $\beta \in [2, 3]$ | $\sigma \in [5, 15]$ $\rho \in [15, 35]$ $\beta \in [1.5, 3.5]$ | 2000 |
| Falling object | NA | $c \in [0.01, 2.0]$ | 1000 |
| Fitzhugh Nagumo | NA | $\epsilon \in [0.005, 0.15]$ $a \in [0.4, 1.2]$ $b \in [0.4, 1.2]$ $I \in [0.3, 2.0]$ | 2000 |
| Pendulum gravity | NA | $l \in [0.5, 2]$ $b \in [0.05, 1]$ | 1000 |
| Driven damped pendulum | NA | $A \in [0.1, 2]$ $b \in [0.1, 2]$ $c \in [1, 5]$ $\omega \in [0.5, 3]$ | 1000 |
| Rössler | NA | $a \in [0.05, 0.35]$ $b \in [0.05, 0.35]$ $c \in [3.0, 10.0]$ | 2000 |

## A.4 NEURVEC IMPLEMENTATION DETAILS

**NeurVec Model Architecture.** The NeurVec model incorporates a multi-layer perceptron (MLP) with a custom activation function, defined as follows:

- **Input layer:** The state dimension of the system .
- **Hidden layer:** 1024 neurons with a custom rational activation function.
- **Output layer:** The error correction term for the state update.

The rational activation function is defined by:

$$f(x) = \frac{a_3 x^3 + a_2 x^2 + a_1 x + a_0}{b_2 x^2 + b_1 x + b_0},$$

where the parameters are: $a_0 = 0.0218, a_1 = 0.5000, a_2 = 1.5957, a_3 = 1.1915, b_0 = 1.0000, b_1 = 0.0000$, and $b_2 = 2.3830$.

**NeurVec Training Details.** We trained the NeurVec model on coarse time step data derived from various dynamical systems. The training parameters were as follows:

- **Learning Rate:** 0.001
- **Optimizer:** Adam
- **Learning Rate Decay:** cosine annealing schedule
- **Batch Size:** 500
- **Number of Epochs:** 100
- **Loss Function:** Mean Squared Error (MSE)

## A.5 TEXTUAL DATA GENERATION

We generated the supplemental textual data for each ODE systems with the assistance of GPT-4. For any ODE systems that we are interested in, we use the following prompt for generation.

Table 9: Performance of FMint on unseen ODEs in RMSE

| Method | # Samples | Unseen ODE | | | | |
| | | Driven damped | Falling | Fitzhugh Nagumo | Pendulum | Rossler |
| --- | --- | --- | --- | --- | --- | --- |
| FMint | 25 | 1.64e-3 | 4.18e-3 | 5.59e-2 | 1.55e-3 | 2.57e-3 |
| | 50 | 9.68e-4 | 5.26e-3 | 5.8e-2 | 1.7e-3 | 2.56e-3 |
| | 200 | 1.34e-3 | 4.74e-3 | 5.3e-2 | 1.6e-3 | 2.5e-3 |
| | 1000 | 9.02e-4 | 3.7e-3 | 4.7e-2 | 1.69e-3 | 2.4e-3 |
| NeurVec | 50000 | 0.0741 | 0.8351 | 0.3859 | 0.0974 | 0.0265 |
| Neural ODE | 50000 | 0.6667 | 1.3754 | 1.2139 | 1.0763 | 0.1571 |

```
Please use [numbers required] different ways to explain what [ODE
system], [mathematical formula] is, where it can be applied and what the
meaning of [parameters].  Also include what types of behavior will there
be for different ranges of its parameters.
```

Take Lorenz attractor as an example, we use the following prompt:

```
Please use 30 different ways to explain what Lorenz attractor,
$\frac{dx}{dt} = \sigma (y - x), \frac{dy}{dt} = x (\rho - z) - y,
\frac{dz}{dt} = xy - \beta$ is, where it can be applied and what the
meaning of $\sigma$, $\rho$ and $\beta$.  Include what types of behavior
will there be for different ranges of its parameters.
```

We have generated 30 textual data for each ODE systems, with slight manual adjustments applied after GPT-4 generation. For pretrained ODEs, we reserved placeholders for the actual parameters used for training and testing. For out-of-distribution ODEs, we did not specify the exact parameters used in our textual data.

During pretraining stage, the textual data is randomly selected from all the provided prompts that correspond to the ODE system. The placeholders for parameters are replaced with the actual parameter values associated with the numerical data. During inference stage, textual data is also randomly selected but only the pretraining ODE systems are filled with exact parameters.

### A.6 TEXTUAL DATA EXAMPLE

Here we show several examples of the textual data for pretraining ODEs and out-of-distribution ODEs: pretraining ODE Lorenz attractor and out-of-distribution ODE Rössler attractor:

- Prompts for Lorenz attractor:
  ```
  It models the evolution of three variables|$x,y,z$ over
  time, governed by three parameters:  $\sigma = 12.69, \rho
  = 34.297, \beta = 2.592$.
  The Lorenz system of equations $\frac{dx}{dt} = \sigma (y
  - x), \frac{dy}{dt} = x (\rho - z) - y, \frac{dz}{dt} =
  xy - \beta$ serves as a classic example in chaos theory,
  showing how a deterministic system can exhibit aperiodic,
  non-repeating, and sensitive trajectories despite being
  governed by simple rules.
  ```

- Prompts for Rössler attractor:
  ```
  The Rössler system $\dot{x} = -y - z, \dot{y} = x + a y,
  \dot{z} = b + z (x - c)$ is a set of three differential
  equations used to model chaotic behavior.  It has parameters
  $a$, $b$ and $c$ which control the system's dynamics.
  When $c$ is low, oscillations are periodic; as $c$
  increases, the system becomes chaotic, which can be used
  to design chaotic communication systems.
  ```

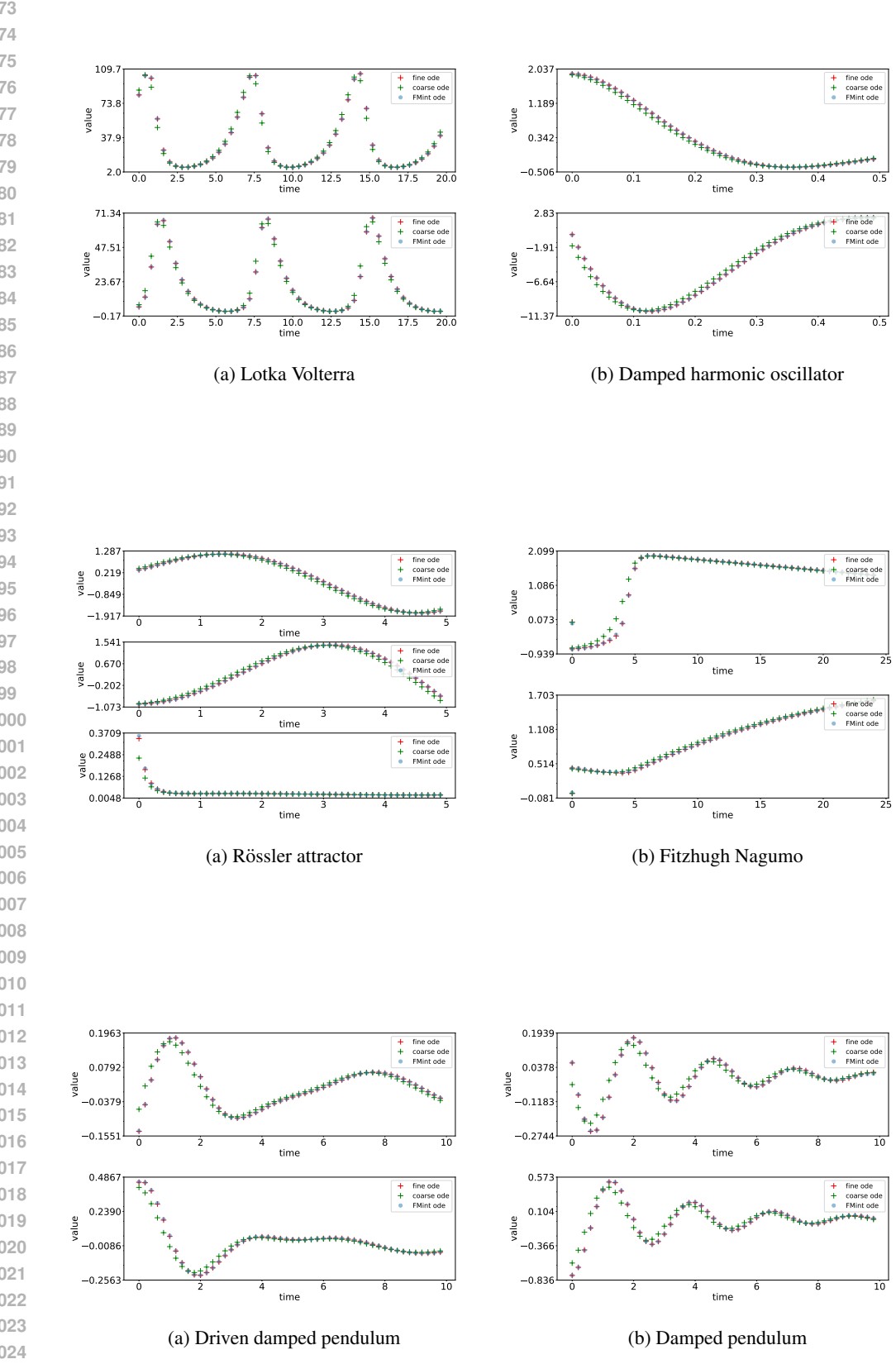

(a) Lotka Volterra

(b) Damped harmonic oscillator

(a) Rössler attractor

(b) Fitzhugh Nagumo

(a) Driven damped pendulum

(b) Damped pendulum

## A.7 VISUALIZATION OF FMINT ON ALL ODES

## A.8 DETAILS ON COEFFICIENT RANGE FOR EACH BEHAVIOR CATEGORY

**Lorenz oscillatory.** $\sigma \in [5, 15], \rho \in [13, 24.74], \beta \in [1.5, 3.5]$.The system exhibits converging oscillatory behavior toward either of the two fixed points.

**Lorenz chaotic.** $\sigma \in [5, 15], \rho \in [24.74, 100], \beta \in [1.5, 3.5]$. This is the famous region where the Lorenz attractor forms, with characteristic of never repeating chaotic motion and sensitive dependence on initial conditions.

**Lorenz chaotic (complex).** $\sigma \in [5, 15], \rho \in [24.74, 150], \beta \in [1.5, 3.5]$. It contains more complicated chaotic dynamics with larger attractor regions and more intricate patterns.

**Driven damped pendulum underdamped.** $b \in [0.01, 0.1], c \in [1.0, 2.0], A \in [0.1, 0.5], \omega \in [0.5, 3.0]$. In this parameter range, the underdamped pendulum may achieve steady oscillations.

**Driven damped pendulum overdamped.** $b \in [0.5, 2.0], c \in [1.0, 2.0], A \in [0.1, 0.5], \omega \in [0.5, 3.0]$. It occurs when the damping is very strong. The pendulum returns to its equilibrium position without oscillating.

**Driven damped pendulum chaotic.** $b \in [0.1, 0.2], c \in [1.0, 2.0], A \in [1, 3.0], \omega \in [0.5, 3.0]$. The chaotic behavior emerges and the system exhibits nonlinear and non-periodic motion. It is highly sensitive to initial conditions.

**FitzHugh-Nagumo resting.** $I \in [0.0, 0.1], \epsilon \in [0.08, 0.1], a \in [0.5, 0.7], b \in [0.1, 0.2]$. In this parameter range, the membrane potential stays close to its equilibrium value.

**FitzHugh-Nagumo spikes.** $I \in [0.1, 0.5], \epsilon \in [0.01, 0.5], a \in [0.7, 1.0], b \in [0.2, 0.25]$. The system can exhibit a single action potential or spike and followed by recovery.

**FitzHugh-Nagumo bursting.** $I \in [0.5, 1.5], \epsilon \in [0.01, 0.02], a \in [0.9, 1.1], b \in [0.15, 0.2]$. In this state, the system may have transition between excitable and oscillatory behavior, with possible periodic spiking.

**FitzHugh-Nagumo oscillatory.** $I \in [0, 2], \epsilon \in [0.01, 0.1], a \in [0.5, 1.2], b \in [0.1, 0.3]$. The system can exhibit bistability, or mixed mode oscillations.

**Rössler periodic.** $a \in [0.1, 0.2], b \in [0.1, 0.2], c \in [4, 5]$. The trajectory in phase space forms a closed loop, and we may see that the system returns to the same state after a fixed period.

**Rössler chaotic.** $a \in [0.2, 0.3], b \in [0.2, 0.25], c \in [5, 9]$. In this state, the system exhibits unpredictable behavior that is sensitive to the initial conditions.

**Rössler hyperchaos.** $a \in [0.25, 0.4], b \in [0.25, 0.3], c \in [9, 13]$. The system is chaotic under this coefficient range in more than one direction, and may have even more complex and unpredictable behavior than standard chaos.

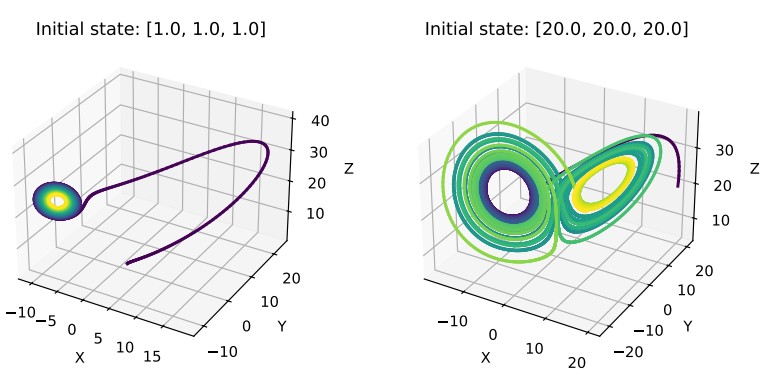

Figure 9: Lorenz attractors - oscillatory. Trajectories generated with $\sigma = 12.69, \beta = 2.59, \rho = 24.33$ under two initial conditions $[1, 1, 1]$ and $[20, 20, 20]$ for 10000 steps with $\Delta t = 0.004$.

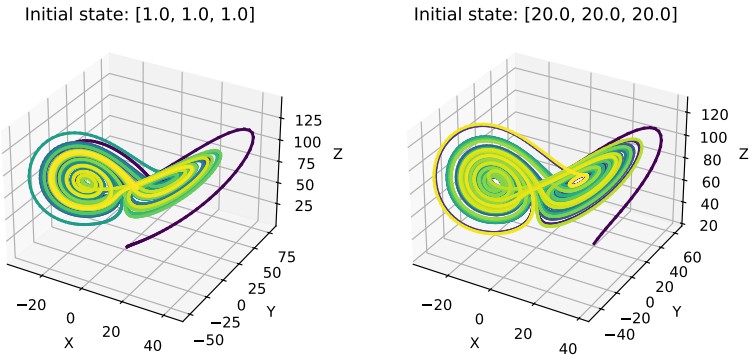

Figure 10: Lorenz attractors - chaotic. Trajectories generated with $\sigma = 12.69, \beta = 2.59, \rho = 78.06$ under two initial conditions $[1, 1, 1]$ and $[20, 20, 20]$ for 10000 steps with $\Delta t = 0.004$.

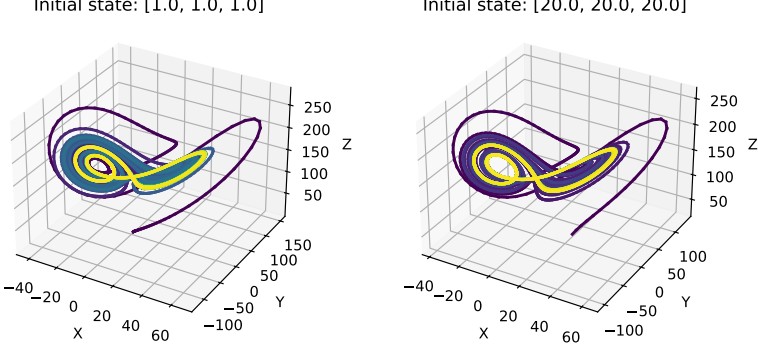

Figure 11: Lorenz attractors - chaotic (complex). Trajectories generated with $\sigma = 12.69, \beta = 2.59, \rho = 78.06$ under two initial conditions $[1, 1, 1]$ and $[20, 20, 20]$ for 10000 steps with $\Delta t = 0.004$.

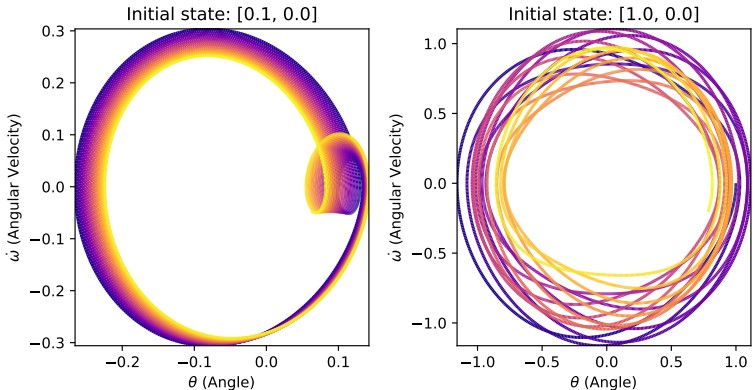

Figure 12: Driven damped pendulum - underdamped. Trajectories generated with $b = 0.005, c = 1.0, A = 0.25, \omega = 2.0$ under two initial conditions $[0.1, 0]$ and $[1.0, 0]$ for 10000 steps with $\Delta t = 0.01$.

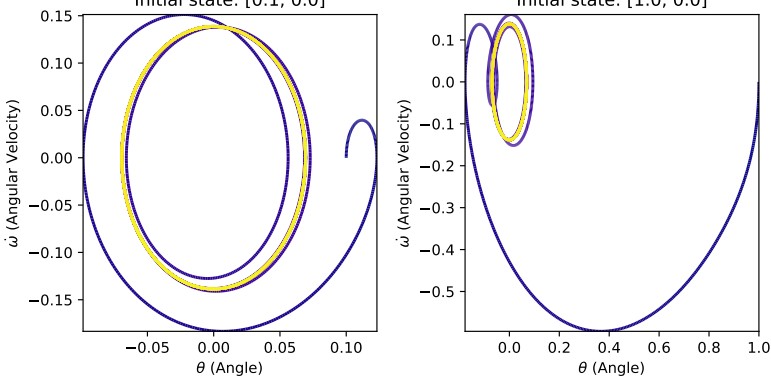

Figure 13: Driven damped pendulum - overdamped. Trajectories generated with $b = 1.0, c = 1.0, A = 0.25, \omega = 2.0$ under two initial conditions $[0.1, 0]$ and $[1.0, 0]$ for 10000 steps with $\Delta t = 0.01$.

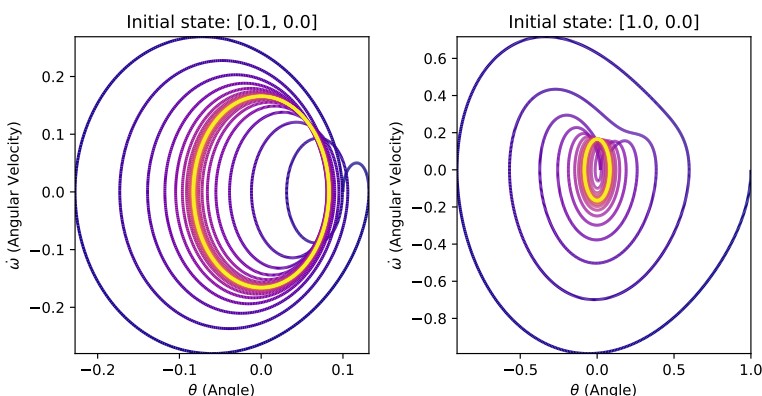

Figure 14: Driven damped pendulum - chaotic. Trajectories generated with $b = 0.15, c = 1.0, A = 0.25, \omega = 2.0$ under two initial conditions $[0.1, 0]$ and $[1.0, 0]$ for 10000 steps with $\Delta t = 0.01$.

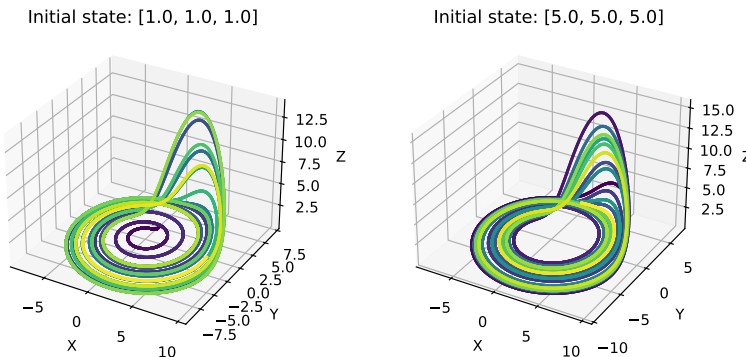

Figure 15: Rössler - periodic. Trajectories generated with $a = 0.179, b = 0.159, c = 4.81$ under two initial conditions $[1, 1, 1]$ and $[5, 5, 5]$ for 10000 steps with $\Delta t = 0.01$.

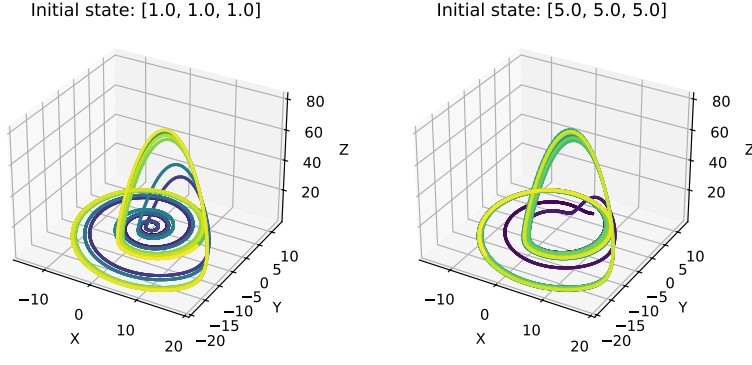

Figure 16: Rössler - chaotic. Trajectories generated with $a = 0.368, b = 0.279, c = 12.24$ under two initial conditions $[1, 1, 1]$ and $[5, 5, 5]$ for 10000 steps with $\Delta t = 0.01$.

