# OpenReview forum: "FMint: Bridging Human Designed and Data Pretrained Models for Differential Equation Foundation Model"
_ICLR.cc/2025/Conference — ICLR 2025 Conference Withdrawn Submission_

### Official Review · Reviewer_zWFN · 2024-10-29

**Soundness:** 3
**Presentation:** 3
**Contribution:** 3
**Rating:** 6
**Confidence:** 3

**Summary:**

This paper proposes a new method for fast simulation of dynamical systems, FMint, which is based on the decoder-only transformer architecture. The framework adopts the context learning method, uses numerical and text data to learn the general error correction scheme of dynamical systems, and uses the prompt sequence of rough solutions from traditional solvers. The method is pre-trained on 400K ODEs. Their results show that FMint improves accuracy by 1-2 orders of magnitude and computation speed compared to traditional numerical solvers.

**Strengths:**

1. Based on transformer architecture and using context learning method
2. The accuracy is obviously improved, which is 1-2 orders of magnitude higher than the existing numerical method
3. High computational efficiency, 5 times faster than traditional numerical methods

**Weaknesses:**

1. Lack of error bound analysis
2. Lack of discussion of long-term predictions for different types of systems

**Questions:**

How to prove that the error correction term can effectively improve the limitations of traditional numerical methods?

---

> ### Author Response · Authors · 2024-11-25
> **Additional questions and comments**
>
> Thanks again for your time and your comments. Is there any remaining concern or question about our paper? We are more than delighted to address any questions you may have.

---

### Official Review · Reviewer_S4yX · 2024-11-03

**Soundness:** 3
**Presentation:** 3
**Contribution:** 3
**Rating:** 6
**Confidence:** 4

**Summary:**

The authors propose a foundation model based upon multi-modal information for time-stepping of dynamical systems.

**Strengths:**

There are some good things to thing to think about here.  I think everyone is interested in good foundation models for dynamical systems.  So the authors have proposed some interesting designs to here build upon.  From that point, I think the paper might be acceptable to ICLR.

**Weaknesses:**

Although the authors claim to "generalize" the method on to new ODES:

"These test ODEs are qualitatively different from the training data, thus convincing to validate FMint’s capability as a foundation model for dynamical systems."

The fact is, these new ODEs have very much the same qualitative behavior as those on which it was trained.  This statement is just false.  Specifically, the "driven-damped pendulum (2D), falling object with air resistence (2D), FitzHugh-Nagumo systems (2D), Pendulum under gravity (2D), and the Ro ̈ssler dynamics (3D)." are very similar to the training data.


A big concern for me:  the testing on training data or against the speed of ODE solvers seems false.  ONE WOULD ALMOST NEVER USE A FIXED TIME STEP!  Instead, everyone uses adaptive stepping which is often included in adaptive stepping RK45.  To fix the time stepper is a completely unfair way to compare to traditional time-steppers.  This raises serious concerns about the comparisons and actual performance claims for this model.

Also see the question below:  I'm not convinced on the performance frankly as I think the generalization data is very similar.  Moreover, I'm not sure they trained in a fair manner.

Also, there is no comparison to ODEFormer:  https://arxiv.org/abs/2310.05573
Which is also a foundation model in this direction.

Also, there is no comparison to broader time-series foundation models or things like S4, Mamba etc.  These also can be used for characterizing the dynamics of these systems which are efficient.

**Questions:**

For clarification, in Sec. 2.1 the data preparation seems to be based upon Euler stepping.  Is this correct?  If so, I find this highly problematics.  One should never use the Euler stepper.... it is highly unstable and very inaccurate.  And adaptive stepping RK45 should be the default.  This detail alone makes me questions the results/performance of the algorithm overall of the paper.  Specifically, I would not trust the results if they are training using Euler.

---

> ### Author Response · Authors · 2024-11-25
> **Additional questions and comments**
>
> Thanks again for your time and your comments. Is there any remaining concern or question about our paper? We are more than delighted to address any questions you may have.

---

### Official Review · Reviewer_KCr5 · 2024-11-04

**Soundness:** 2
**Presentation:** 1
**Contribution:** 2
**Rating:** 3
**Confidence:** 4

**Summary:**

For the simulation of dynamical systems, the paper proposes a multi-modal model that processes both the coarse-grained trajectories and textual data describing the system generated by ChatGPT to obtain fine-grained trajectories. The model is tested on various dynamical systems including Lorenz attractors, driven damped pendulum, FitzHugh-Nagumo and Rössler systems.

**Strengths:**

The idea of integrating textual information with numerical coarse-grained trajectories to simulate dynamical systems is definitely interesting.

**Weaknesses:**

- the authors make strong claims that are hardly substantiated in the paper, in particular the comparison in terms of accuracy is hardly a fair comparison
	- l103: "the first multi-modal foundation model that synthesizes human-designed algorithms and deep learning framework" is too generic and vague, since "human-designed" covers both the standard numerical solvers and machine-learning models
	- l106: the better accuracy, is not correctly substantiated by Table 2. as explained in the following section "Questions"
- the paper contains numerous overstatements and confusing sentences, making it difficult to understand the core arguments. The paper contains too many generic statements in the flavor of arguing that the method is in general "better and faster", see the minor points below

Minor comments
- l025: "corpus" is used for textual data, but I guess you also refer to numerical data in that sentence
- l027: "that exhibit ... high dimensionality" could be better written
- l047: what is the meaning of "format" in that context?
- l053: "human-designed algorithm" any model you mention in the paper is "human-designed", this expression does not add anything
- l054-055: there is a repetition "real-world scenarios" "various applications"
- l068-069: the question is not well formulated. For many equations, the classical solvers appear to be actually both more accurate and more efficient than their neural network counterpart. However, neural network used to solving ODEs are certainly more flexible and allow for generalization to unseen ODEs to a certain extent.
- l070-071: repetition "foundation" "foundation"
- l135-136: this sentence is misleading since the Vaswani et al. paper uses an encoder-decoder and not a "decoder-only" architecture
- l247: "baslines"
- l248: "research questions" should exist independently of your paper. However, the three research questions are related to your model
- l322: "higher than", do you mean "lower than"?
- l378: table 3. The title is misleading. You mention "unseen ODEs" while doing fine-tuning, so actually "seeing" these ODEs.

**Questions:**

- Table 2. Most of the baselines scores are very bad (>1). Is it because such models should not be used for such tasks? Or is it because the training didn't go well in your experiments?
- Table 2. I am having a hard time convincing myself that the comparison with the baselines is fair. Does the baseline take as input the coarse-grained trajectory as well? It seems to me that this is not the case (take for example neuralODE). Then, I am not surprised that a network which takes as an input the coarse-grained trajectory is performing better.

---

> ### Comment · Reviewer_KCr5 · 2024-11-12
>
> Dear authors and AC,
> With the release of the reviews, I realized that I mistakenly swapped two of my reviews. The review I originally uploaded for this paper actually belonged to another paper.
> I apologize sincerely to the authors for this issue.
> I have uploaded the correct review.
> Best regards,

---

> ### Author Response · Authors · 2024-11-25
> **Additional questions and comments**
>
> Thanks again for your time and your comments. Is there anything else we can improve to make our submission more worthy of acceptance? We are more than delighted to address any questions you may have.

---

> > ### Comment · Reviewer_KCr5 · 2024-12-02
> >
> > I thank the author for their response to my points. Unfortunately, there are still important aspects that remain unclear and need improvement. As such, I will maintain my score.
> >
> > One significant point to address is the comparisons in Table 2. The baseline scores are so poor that it is unclear whether they can genuinely be considered a "baseline." To clarify, I am not questioning the suitability of these models for the task but rather the appropriateness of the trained versions you used. If the baselines were indeed correctly trained, I suggest including an additional experiment or providing a theoretical argument to demonstrate that these baselines cannot perform well in your specific case. The aim is to justify that the very high MAE and RMSE scores for the baselines are valid and reflective of their limitations.

---

### Official Review · Reviewer_SRVs · 2024-11-06

**Soundness:** 3
**Presentation:** 3
**Contribution:** 2
**Rating:** 3
**Confidence:** 2

**Summary:**

The paper presents FMint, a multi-modal foundation model for fast simulation of dynamical systems. It combines human-designed and data-driven approaches using a decoder-only transformer architecture with in-context learning, leveraging both numerical and textual data for error correction. Pre-trained on 40,000 ordinary differential equations (ODEs), FMint demonstrates significant improvements in accuracy and efficiency, achieving 1 to 2 orders of magnitude better accuracy and a 5X speedup over traditional numerical solvers, making it a promising general-purpose solver for dynamical systems.

**Strengths:**

1.The FMint model uniquely combines numerical and textual data, enhancing its robustness in diverse ODE environments

2.The use of in-context learning for error correction in dynamical systems showcases a novel integration of human-designed and data-driven algorithms, enhancing the model’s ability to handle data-scarce environments

**Weaknesses:**

1.Some details in the paper are insufficiently described. For example, the length of the data trajectories and the number of pre-training contexts are not specified. These details are crucial to the experimental design as they relate to the transformer’s input length and memory usage.

2.While the model benefits from multi-modal input, its performance may decrease in the absence of textual data, especially for complex systems. How does the model perform on high-dimensional ODEs with limited numerical data and no textual data?

3.Since this is in-context learning, it would be beneficial to conduct few-shot experiments without fine-tuning and to examine how context length impacts results.

**Questions:**

1.The details of the natural language prompt are lacking. What kind of embedding was used?

2.A plausible hypothesis is that the language prompt mainly helps the model differentiate between system types. Without it, systems like LV and the Van der Pol oscillator, which have similar shapes, might be confused with each other if only numeric prompts are used. A useful experiment might be to use simple category prompts (e.g., one for each system type) as a control, to demonstrate that complex natural language prompts indeed add value and that the model extracts meaningful information from them.

3.Can this model truly leverage information as specific as the symbolic form of the system’s control equations? Are there further experiments to support this?

4.I suggest including the MAE for a coarse prediction as a comparison.

5.Transformers are powerful but computationally intensive. Has there been any consideration of alternative architectures that could achieve similar results with lower computational overhead?

6.FMint’s performance is primarily compared to Neural ODE, NeurVec, and ICON-LM. Could additional benchmarks, like recent neural operators, provide further insights into its relative strengths and limitations?

---

> ### Author Response · Authors · 2024-11-25
> **Additional questions and comments**
>
> Thanks again for your time and your comments. Is there anything else we can improve to make our submission more worthy of acceptance? We are more than delighted to address any questions you may have.

---

### Note · Authors · 2024-12-02

I have read and agree with the venue's withdrawal policy on behalf of myself and my co-authors.